# Diverse Policies Recovering via Pointwise Mutual Information Weighted Imitation Learning

**Hanlin Yang**$^{12*\dagger}$**, Jian Yao**$^{2*}$**, Weiming Liu**$^2$**, Qing Wang**$^2$**, Hanmin Qin**$^2$**, Hansheng Kong**$^2$**,**
**Kirk Tang**$^2$**, Jiechao Xiong**$^2$**, Chao Yu**$^{1\ddagger}$**, Kai Li**$^3$**, Junliang Xing**$^4$**, Hongwu Chen**$^2$**,**
**Juchao Zhuo**$^2$**, Qiang Fu**$^2$**, Wei Yang**$^2$**, Haobo Fu**$^{2\ddagger}$

$^1$ Sun Yat-sen University, Guangzhou, China
$^2$ Tencent AI Lab, Shenzhen, China
$^3$ Institute of Automation, Chinese Academy of Sciences, Beijing, China
$^4$ Tsinghua University, Beijing, China

## Abstract

Recovering a spectrum of diverse policies from a set of expert trajectories is an important research topic in imitation learning. After determining a latent style for a trajectory, previous diverse policies recovering methods usually employ a vanilla behavioral cloning learning objective conditioned on the latent style, treating each state-action pair in the trajectory with equal importance. Based on an observation that in many scenarios, behavioral styles are often highly relevant with only a subset of state-action pairs, this paper presents a new principled method in diverse polices recovery. In particular, after inferring or assigning a latent style for a trajectory, we enhance the vanilla behavioral cloning by incorporating a weighting mechanism based on *pointwise mutual information*. This additional weighting reflects the significance of each state-action pair's contribution to learning the style, thus allowing our method to focus on state-action pairs most representative of that style. We provide theoretical justifications for our new objective, and extensive empirical evaluations confirm the effectiveness of our method in recovering diverse policies from expert data.

## 1 Introduction

Imitation Learning (IL) is about observing expert demonstrations in performing a task and learning to mimic those actions (Hussein et al., 2017; Osa et al., 2018). Vanilla behavioral cloning (BC) (Pomerleau, 1991) learns a mapping from state to actions using expert state-action pairs via supervised learning, which is simple to implement but may have the issue of compounding errors. Generative Adversarial Imitation Learning (GAIL) (Ho & Ermon, 2016) mitigates the issue via learning both a discriminator and a policy. Despite their wide applications, these methods lack mechanisms to generate diverse policies, which may be essential in certain tasks.

There has been an increase in recent research addressing policy diversity in imitation learning (Li et al., 2017; Wang et al., 2017; Zhan et al., 2020; Shafiullah et al., 2022; Mao et al., 2023), which can generally divided into two categories. In one category, the latent style $z$ of a trajectory is inferred in an unsupervised manner, for instance, by an expectation maximization (EM) procedure. In the other category, the style $z$ of a trajectory is determined by a user-specified function, for instance, a programmatic labeling function. No matter in which category, samples that are used to train a style-conditioned policy $\pi(a|s,z)$ are treated with equal importance in those methods. However, in many cases, we observe that the relevance of different state-action pairs to the trajectory style can vary significantly. For example, in autonomous driving tasks, the diversity of overtaking policies (from the left or right) is primarily relevant to the overtaking period in the trajectory. In other words, the preceding normal driving period is less relevant to the overtaking diversity. Again, in a basketball game, the diversity of shooting position behaviors is primarily related to the part leading up to the

---

$^*$Equal contribution
$^\dagger$Work done during an internship at Tencent AI Lab
$^\ddagger$Equal advising. Correspondence to: Haobo Fu (haobofu@tencent.com).

shot, while it is less relevant to other parts in the trajectory. In a nutshell, policies with different styles could have significantly different degrees of overlap in different areas of the state-action space. Therefore, when learning a style-conditioned policy, the relevance between the state-action pairs in the trajectory and the behavioral style should be taken into consideration.

In this paper, we propose a new diverse policies recovering method by leveraging the relevance of the state-action pairs with the trajectory styles. We focus on the situation where the style label of a trajectory has been provided, as unsupervised learning (either by EM or mutual information maximization) of the style $z$ leads to uncontrolled styles and often only qualitative evaluation. In particular, we introduce an additional importance weight based on Pointwise Mutual Information (PMI) (Church & Hanks, 1990; Manning & Schutze, 1999; Bouma, 2009), in traditional conditional BC, to quantify the relevance of state-action pairs with the conditioned style. Intuitively, state-action pairs with a larger posterior of the corresponding style $p(z|s, a)$ are given a larger weight. In practical implementation, we utilize Mutual Information Neural Estimation (MINE) (Belghazi et al., 2018) to estimate the PMI between state-action pairs and style variables. We term the proposed method Behavioral Cloning with Pointwise Mutual Information Weighting (BC-PMI).

Our theoretical analysis indicates that our new weighted learning objective unifies two extreme cases in recovering diverse policies. When the mutual information between the style and the state-action pair is zero, which means there is no style diversity in expert data, our objective degenerates to vanilla BC, which views the data as generated from one policy. By contrast, when policies with different styles have no overlap in the state-action space, our objective degenerates to learning different style polices separately. Empirical results in Circle 2D, Atari games and professional basketball player dataset demonstrate that BC-PMI achieves better performance in recovering diverse policies than the baseline methods.

## 2 RELATED WORK

**Imitation Learning**    Imitation learning (IL) methods are designed to mimic the behaviors of experts. Behavioral Cloning (BC) (Pomerleau, 1991), a well-known IL algorithm, learns a policy by directly minimizing the discrepancy between the agent and the expert in the demonstration data. However, offline learning methods like BC suffer from compounding errors and the inability to handle distributional shifts during evaluation (Ross et al., 2011; Fujimoto et al., 2019; Wu et al., 2019; Peng et al., 2019; Kostrikov et al., 2021). Inverse Reinforcement Learning (IRL) (Ng et al., 2000; Arora & Doshi, 2021), another type of IL, learns a reward function that explains the expert behavior and then uses this reward function to guide the agent's learning process. Popular IRL approaches like Generative Adversarial Imitation Learning (GAIL) (Ho & Ermon, 2016) and Adversarial Inverse Reinforcement Learning (AIRL) (Fu et al., 2017) use adversarial training to learn a policy that is similar to the expert policy while being robust to distributional shift. However, these methods are limited to imitating a single policy and do not address the issue of promoting diverse policies. When imitating diverse policies, BC approaches using supervised learning tend to learn an average policy, which does not fully capture the range of diverse behaviors (Codevilla et al., 2018). GAIL tends to learn a policy that captures only a subset of the expert's control behaviors, which can be viewed as modes of distribution. Consequently, the learned policy fails to cover all styles of the expert's diverse behaviors Wang et al. (2017).

**Policy Diversity**    Diversity is crucial in imitation learning algorithms, especially in practical control tasks and multi-player games (Zhu et al., 2018), as diverse policies in control tasks can enhance the robustness of adapting to various environments. In contrast, AI with diverse policies can maximize the player's experience and the ornamental value in games and competitions (Yannakakis & Togelius, 2018). Some approaches utilize information-theoretic methods to address this issue and learn the behavioral styles of policies. InfoGAIL (Li et al., 2017) and Intention-GAN (Hausman et al., 2017) augment the objective of GAIL with the mutual information between generated trajectories and the corresponding latent codes. Wang et al. (2017) use a variational autoencoder (VAE) module to encode expert trajectories into a continuous latent variable. Eysenbach et al. (2018) propose a general method called Diversity Is All You Need (DIAYN) for learning diverse skills without explicit reward functions. DIAYN focuses on discovering skills in online reinforcement learning tasks (Campos et al., 2020; Sharma et al., 2019; Achiam et al., 2018), whereas our method specifically considers pure offline imitation learning scenarios. More recently, Mao et al. (2023) introduced

Stylized Offline RL (SORL), which utilizes unsupervised learning methods to cluster styles and fits policies to each cluster separately, optimizing an equal number of policies while using the KL divergence as a constraint. However, these methods focus on inferring the latent style or clustering the trajectories with different styles without considering the relevance between the state-action pairs in the trajectory and the behavioral style. In contrast, we focus on the situation where the style label of a trajectory has been provided. By introducing the PMI weights, we quantify the relevance of state-action pairs with the conditioned style, allowing the policy to focus on the samples that are highly relevant to the style.

## 3 PRELIMINARY

### 3.1 PROBLEM SETTING AND VANILLA BEHAVIOR CLONING

We consider the standard Markov Decision Process (MDP) (Sutton & Barto, 2018) as the mathematical framework for modeling sequential decision-making problems, which is defined by a tuple $\langle \mathcal{S}, \mathcal{A}, P, r, d_0, T \rangle$, where $\mathcal{S}$ is a finite set of states, $\mathcal{A}$ is a finite set of actions, $P : \mathcal{S} \times \mathcal{A} \times \mathcal{S} \to \mathbb{R}$ is the transition probability function, $r : \mathcal{S} \to \mathbb{R}$ is the reward function, $d_0$ is the initial distribution, and $T$ is an episode horizon. A policy $\pi : \mathcal{S} \times \mathcal{A} \to [0, 1]$ maps from state to distribution over actions. Let $d_t^\pi$ and $d^\pi = \frac{1}{T} \sum_{t=1}^{T} d_t^\pi$ denote the distribution over states at time step $t$ and the average distribution over $T$ time steps induced by $\pi$, respectively. The vanilla BC loss function is:

$$\mathcal{L}_{\text{BC}} = \mathop{\mathbb{E}}_{(s,a) \sim \mathcal{D}_e} \big[ -\log \pi(a|s) \big]. \tag{1}$$

which aims to maximize the probability of selecting action $a$ for a given policy under the state $s$.

Building upon the basic setup, we focus on an assumption where the expert demonstrations $\mathcal{D}_e$, which consist of many diverse trajectories $\tau$, are collected by stylized expert policies denoted as $\{\pi_e^{(1)}, \pi_e^{(2)}, \ldots, \pi_e^{(K)}\}$. Let $z \in \mathcal{Z}$ denotes the variable indicating which stylized policy $\tau$ belongs to, and $p(\tau|z = i)$ denotes the probability of $\tau$ sampled under policy $\pi_e^{(i)}$. Our objective is to learn a conditioned policy $\pi(a|s, z)$ such that trajectories generated by $\pi(a|s, z)$ closely match the demonstrations in $\mathcal{D}_e$ that exhibit the corresponding style $z$.

### 3.2 MUTUAL INFORMATION NEURAL ESTIMATION

Mutual Information Neural Estimation (MINE) is a powerful technique for estimating the mutual information between two random variables using neural networks (Belghazi et al., 2018). It has been widely used in various domains, including representation learning (Hjelm et al., 2018), generative modeling (Chen et al., 2016), and imitation learning (Eysenbach et al., 2018). The key idea behind MINE is to formulate the estimation of mutual information as an optimization problem. Given two random variables $X$ and $Y$, the mutual information $I(X; Y)$ can be expressed as the Kullback-Leibler (KL) divergence between the joint distribution $P_{XY}$ and the product of marginal distributions $P_X \otimes P_Y$:

$$I(X; Y) = D_{KL}(P_{XY} || P_X \otimes P_Y). \tag{2}$$

MINE approximates this KL divergence using a lower bound based on the Donsker-Varadhan representation (Donsker & Varadhan, 1983):

$$I(X; Y) \geq \sup_{T_\theta \in \mathcal{F}} \mathbb{E}_{P_{XY}}[T_\theta] - \log(\mathbb{E}_{P_X \otimes P_Y}[e^{T_\theta}]), \tag{3}$$

where $\mathcal{F}$ is a class of functions $T : \mathcal{X} \times \mathcal{Y} \to \mathbb{R}$. In MINE, this function class is parameterized by a neural network $T_\theta$, which takes as input samples from the joint distribution $P_{XY}$ and the product of marginal distributions $P_X \otimes P_Y$. The network is trained to maximize the lower bound, equivalent to estimating the mutual information.

## 4 BEHAVIORAL CLONING WITH POINTWISE MUTUAL INFORMATION WEIGHTING

As discussed in the introduction, in many real-world applications, the impact of different state-action pairs on the style can vary greatly, and often only a part of the trajectory is highly relevant to the

style. Hence, when training a style-conditioned policy, we would like to assign different weights to different state-action pairs based on their relevance to that style. We aim to develop such a method that can capture the specific influence of each $(s, a)$ pair on the style, thereby achieving a more generalized imitation objective:

$$\min_{\pi} \mathbb{E}_{(s,a,z)\sim\mathcal{D}_e} \left[ -\log \pi(a|s,z) \cdot \sigma(s,a,z) \right], \tag{4}$$

where $\sigma(s, a, z)$ is a weighting function. Intuitively, state-action pairs that are more exclusive to a style should have larger weights in learning that style-conditioned policy, and vice-versa. Drawing inspiration from the information theory, we introduce the PMI (Church & Hanks, 1990; Manning & Schutze, 1999; Bouma, 2009) to quantify the contribution of $(s, a)$ when learning a style-conditioned policy:

$$\mathcal{P}(z; s, a) = \log \frac{p(s, a, z)}{p(s, a)p(z)} = \log \frac{p(z|s, a)}{p(z)}. \tag{5}$$

We utilize the exponential of $\mathcal{P}(z; s, a)$ as the weight, which is the ratio of the posterior given $(s, a)$ and the prior of the style $z$. This weight has the following properties. When the posterior of style $z$ given $(s, a)$ is larger than the prior, i.e., $p(z|s, a) > p(z)$, this means there is a high relevance between $(s, a)$ and style $z$, and we should give a larger weight. If $(s, a)$ and style $z$ are nearly independent, then $p(z|s, a) \approx p(z)$. When we have $p(z|s, a) < p(z)$, this means (s,a) is more likely generated by other styles, and we should give a lower weight here. Accordingly, Eq.(4) turns into the following form:

$$\mathcal{L}_{\mathrm{BC-PMI}}(\theta) = \mathbb{E}_{(s,a,z)\sim\mathcal{D}_e} \left[ -\log \pi_\theta(a|s, z) \cdot e^{\mathcal{P}(z;s,a)} \right] = \mathbb{E}_{(s,a,z)\sim\mathcal{D}_e} \left[ -\log \pi_\theta(a|s, z) \cdot \frac{p(z|s, a)}{p(z)} \right] \tag{6}$$

## 4.1 THEORETICAL ANALYSIS

We give some theoretical justifications for our BC-MI objective in Eq.(6). Considering an extreme case where the mutual information between $(s, a)$ and style $z$ approaches zero, which means policies with different styles have nearly the same state-action distribution. In this case, the BC-PMI objective degenerates into vanilla BC, which means there is no difference in training style conditioned policies and an average unconditioned policy. In the opposite extreme case, where all $(s, a)$ pairs exhibit significant style differences, which means trajectories of different styles have minimal overlap, we find that the BC-PMI objective degenerates into behavior cloning on each style. Formally, we have the following proposition:

**Proposition 1.** *(a). When the mutual information $I(Z; S, A)$ equals to $0$, it indicates that there is no distinction in the trajectory style corresponding to all the state-action pairs. In this case, the BC-PMI objective degenerates to the vanilla behavior cloning objective:*

$$\arg\min_\theta \mathcal{L}_{BC\text{-}PMI}(\theta) = \arg\min_\theta \mathcal{L}_{BC}(\theta). \tag{7}$$

*(b). When the conditional entropy $H(Z|S, A)$ equals $0$, it indicates that there is a significant distinction in the trajectory style corresponding to all the state-action pairs. In this case, the BC-PMI objective degenerates to the behavior cloning on each style:*

$$\mathcal{L}_{BC\text{-}PMI}(\theta) = \sum_{i=1}^{K} \mathcal{L}_{\mathrm{BC}}^{(i)}(\theta), \tag{8}$$

*where $\mathcal{L}_{\mathrm{BC}}^{(i)}(\theta)$ is the behavior cloning loss on the subset of data with style label $i$.*

*Proof.* Refer to Appendix A.1.

Equation 8 means when $H(Z|S, A)$ equals to $0$, the BC-PMI objection function can be viewed as separating the $(s, a)$ by the style $z$ and optimize it respectively. This is associated with the clustering-based behavior cloning objective, where each trajectory is first assigned to a specific style cluster based on its style label, and then behavior cloning is performed within each cluster.

The above analysis provides insights into the mechanisms of BC-PMI. In practice, the mutual information $I(Z; S, A)$ often lies between the two extremes of $0$ and $H(Z)$. By adjusting the weight of a state-action sample in learning a style-conditioned policy, BC-PMI can smoothly interpolate between vanilla behavior cloning and clustering behavior cloning, allowing it to use the expert data more effectively than previous methods that treat each state-action sample with equal importance.

## 4.2 PRACTICAL IMPLEMENTATION

To practically estimate the PMI values in Eq.(6), we employ the Mutual Information Neural Estimation (MINE) (Belghazi et al., 2018) method. MINE is a neural network-based approach that can effectively estimate mutual information between high-dimensional random variables. By leveraging the Donsker-Varadhan representation (Donsker & Varadhan, 1983) of the Kullback-Leibler (KL) divergence, MINE allows for the estimation of mutual information using neural networks.

Let $T_\phi : \mathcal{S} \times \mathcal{A} \times \mathcal{Z} \to \mathbb{R}$ be a neural network parameterized by $\phi$. The mutual information between $(s, a)$ and $z$ can be estimated as:

$$\mathcal{I}(s, a; z) \geq \mathbb{E}_{(s,a,z)\sim\mathcal{D}_e} [T_\phi(s, a, z)] - \log \left[ \mathbb{E}_{(s,a)\sim\mathcal{D}_e, \bar{z}\sim p(z)} \left[ e^{T_\phi(s,a,\bar{z})} \right] \right]. \tag{9}$$

The neural network $T_\phi$ is trained to maximize the lower bound in Eq.(9), which is equivalent to minimizing the KL divergence between the joint distribution $p(s, a, z)$ and the product of marginals $p(s, a)p(z)$. According to the proof of Theorem 1 in MINE (Belghazi et al., 2018), the optimal solution for $T_\phi$ is:

$$T_\phi^*(s, a, z) = \log \frac{p(s, a, z)}{p(s, a)p(z)} = \log \frac{p(z|s, a)}{p(z)}, \tag{10}$$

which is exactly the PMI we aim to estimate.

In practice, we can train the MINE network $T_\phi$ using samples from the expert demonstrations $\mathcal{D}_e$ and the style distribution $p(z)$. The training objective for $T_\phi$ is:

$$\max_\phi \mathbb{E}_{(s,a,z)\sim\mathcal{D}_e} [T_\phi(s, a, z)] - \log \left[ \mathbb{E}_{(s,a)\sim\mathcal{D}_e, \bar{z}\sim p(z)} \left[ e^{T_\phi(s,a,\bar{z})} \right] \right]. \tag{11}$$

By optimizing Eq.(11), we obtain an approximation of the PMI values, which can be used as the weights in the behavioral cloning objective in Eq.(6). The weights can be denoted as:

$$\sigma(s, a, z) = \exp[T_\phi^*(s, a, z)]. \tag{12}$$

Furthermore, in order to reduce the variance of the gradient in optimizing Eq.(6), we can subtract an optimal baseline from the weight, like what has been done in A3C (Mnih et al., 2016). This turns Eq.(6) to the following objective:

$$\min_\pi \mathbb{E}_{(s,a,z)\sim\mathcal{D}_e} \left[ -\log \pi(a|s, z) \cdot \left[ \exp(T_\phi^*(s, a, z)) - \widetilde{b} \right] \right], \tag{13}$$

where $\widetilde{b} = \mathbb{E}_{(s,a,z)\sim\mathcal{D}_e} \left[ \exp(T_\phi^*(s, a, z)) \right]$. In practice, we can estimate $\widetilde{b}$ using the moving average. The pseudo-code for the BC-PMI algorithm is shown in Algorithm 1.

## 5 EXPERIMENTS

In this experimental section, we aim to address the following questions:

Q1. Can our method recover diverse and controllable policies from diverse style trajectories?

Q2. Does the PMI weight have interpretability and improve policy diversity and style calibration to the algorithm?

Q3. Can our method cope with complex, real-world tasks, particularly those that involve extensive datasets derived from human participants?

---

**Algorithm 1** Behavioral Cloning with Pointwise Mutual Information Weighting (BC-PMI)

---

Initial the parameters $\phi$ of the neural network $T$, the parameters $\theta$ of the policy $\pi$;
Given the expert demonstrations $\mathcal{D}_e$;
**for** $i = 0, 1, 2, \ldots$ **do**
  Draw $b$ minibatch samples from the joint distribution: $(\{s, a\}^{(1)}, z^{(1)}), \ldots, (\{s, a\}^{(b)}, z^{(b)}) \sim \mathcal{D}_e$;
  Draw $b$ samples from the $Z$ marginal distribution: $\bar{z}^{(1)}, \ldots, \bar{z}^{(b)} \sim P(Z)$;
  Update the neural network $T$ using:

$$\max_{\phi} \frac{1}{b} \sum_{i=1}^{b} T_\phi(\{s, a\}^{(i)}, z^{(i)}) - \log(\frac{1}{b} \sum_{i=1}^{b} e^{T_\phi(\{(s,a)\}^{(i)}, \bar{z}^{(i)})}). \tag{14}$$

**end for**
**for** $i = 0, 1, 2, \ldots$ **do**
  Sample a random minibatch of $N$ state-action pairs from $\mathcal{D}_e$;
  Update the policy $\pi$ with the PMI weighting using:

$$\min_{\pi} \mathop{\mathbb{E}}_{(s,a,z) \sim \mathcal{D}_e} \Big[ -\log \pi(a|s, z) \cdot \big[ \exp(T_\phi^*(s, a, z)) - \widetilde{b} \big] \Big]. \tag{15}$$

**end for**

---

## 5.1 STYLES AND BASELINES

The dimension of the style needs to be specified if the style needs to be controllable. Our method can handle both style-labeled data and data without style labels. For the latter, getting the style labels can be expensive when relying on manual annotations and uncontrollable when using unsupervised approaches. Instead, the programmable labeling functions (Ratner et al., 2016; Zhan et al., 2020) can be used to automatically generate style labels.

We compare PMI-BC to the following baselines: (1) **BC**, which directly imitates expert actions across all styles; (2) **CBC**, which separates trajectories of different styles and uses BC to imitate each style separately; (3) **CTVAE**, which is the conditional version of TVAEs (Wang et al., 2017); (4) **InfoGAIL** (Li et al., 2017), which infers the latent style of trajectories by maximizing the mutual information between the latent codes and trajectories; (5) **SORL** (Mao et al., 2023)*, which use Expectation-Maximization (EM) algorithm to classify the trajectories from the heterogeneous dataset into clusters where each represents a distinct and dominant motion style. Since CBC, CT-VAE, and our method use the style labels, we concatenate the state and style as input to the network of InfoGAIL and SORL to ensure a fair comparison. Specifically, we first introduce a toy example called Circle 2D to provide a simple visualization and analysis. Then in Atari games, detailed analysis and validation were provided regarding the PMI weights. Lastly, we evaluate all these baseline methods in Section 5.4 to illustrate the effectiveness of BC-PMI.

## 5.2 CIRCLE 2D

The Circle 2D environment is a 2D plane where an agent can freely move at a constant velocity by selecting its direction, denoted as $p_t$, at time step $t$. For the agent, the observation at time step $t$ includes the state information from time step $t - 4$ to $t$. The offline expert trajectories consist of four different styles, each generated by a random expert policy. The expert policy generates trajectories that resemble circular patterns after a period of translation (75 time steps). This design aims to introduce partial diversity in the trajectories. In this environment, each episode consists of 300 time steps. If the first loop around the circle is completed before reaching 300 steps, the agent continues circling until the end of the episode. Hence, in this scenario, there is minimal difference in the offline trajectories during the first 75 steps, and the trajectory differences vary as the agent's position progresses after 75 steps. During the imitation learning training process, the expert trajectories used are noisy, meaning there is randomness introduced in both the sampled actions and the environment.

---

*For the SORL algorithm, we only used the EM clustering part.

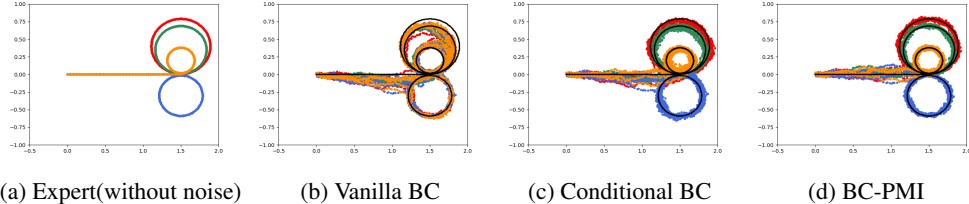

| (a) Expert(without noise) | (b) Vanilla BC | (c) Conditional BC | (d) BC-PMI |

Figure 1: Visualized comparison of trajectories generated by different policies.

This task is a toy example that can help the readers intuitively understand our motivation. In this task, the agent's behavioral diversity is highly related to the curvature of the circle moving after 75 steps and has nothing to do with the linear movement part. Therefore, $(s, a)$ that moves according to the curvature of a specific style after 75 steps will be assigned a higher PMI, and the linear motion part or the movement curvature which are not related to the style will be assigned a lower PMI. The visualized results are shown in Figure 1. By introducing PMI, the calibration of the BC policies has been further improved. Table 1 is a comparison of numerical results. The metrics used are as follows: (1)**DTW**, which is Dynamic Time Warping, a dynamic programming algorithm for calculating the similarity of sequences of different lengths; (2)**ED**, which is Euclidean Distance; (3)**KL**, which is used to calculate the state-action distribution difference of the trajectory.

Table 1: Comparison of style calibration across different metrics in Circle 2D.

| Style | Metrics | BC | CBC | BC-PMI |
|---|---|---|---|---|
| Class 1 (Red) | DTW | $88.590 \pm 9.329$ | $8.130 \pm 0.732$ | $\mathbf{7.511 \pm 0.545}$ |
| | ED | $133.100 \pm 20.210$ | $22.039 \pm 1.383$ | $\mathbf{21.591 \pm 1.091}$ |
| | KL | $8.037 \pm 0.870$ | $0.044 \pm 0.006$ | $\mathbf{0.037 \pm 0.004}$ |
| Class 2 (Green) | DTW | $45.729 \pm 5.293$ | $7.695 \pm 0.814$ | $\mathbf{7.631 \pm 0.821}$ |
| | ED | $88.180 \pm 7.541$ | $\mathbf{35.355 \pm 2.691}$ | $35.777 \pm 2.912$ |
| | KL | $1.111 \pm 0.072$ | $0.043 \pm 0.003$ | $\mathbf{0.037 \pm 0.003}$ |
| Class 3 (Orange) | DTW | $77.619 \pm 7.812$ | $18.972 \pm 2.323$ | $\mathbf{11.576 \pm 1.788}$ |
| | ED | $130.839 \pm 10.322$ | $107.833 \pm 8.213$ | $\mathbf{74.267 \pm 8.002}$ |
| | KL | $16.839 \pm 1.022$ | $0.271 \pm 0.031$ | $\mathbf{0.135 \pm 0.014}$ |
| Class 4 (Blue) | DTW | $69.527 \pm 10.924$ | $7.755 \pm 1.832$ | $\mathbf{7.577 \pm 1.05}1$ |
| | ED | $97.066 \pm 14.239$ | $27.230 \pm 2.110$ | $\mathbf{26.749 \pm 2.347}$ |
| | KL | $36.238 \pm 2.981$ | $0.401 \pm 0.061$ | $\mathbf{0.219 \pm 0.019}$ |

## 5.3 ATARI GAMES

In this experiment, we concentrate on three widely recognized Atari games: Alien, MsPacman and SpaceInvaders. The datasets utilized in this study are sourced from Atari-Head (Zhang et al., 2018; 2020), an extensive collection of human game-play data. Atari-Head are meticulously recorded in a semi-frame-by-frame manner, ensuring high data quality and granularity, which facilitates in-depth analysis and robust evaluation of our proposed method.

This experiment consists of three parts. Firstly, we demonstrate the convergence of the lower bound of mutual information in Eq.(9), which indicates the relevance between state-action pairs and styles. Secondly, we provide interpretability of PMI weights by assessing the extent to which they appropriately reflect the influence of the current $(s, a)$ pair on the style. Lastly, we evaluate the controllability of the BC-PMI policy, which refers to the ability of the policy to act according to a given style once it is specified. In Atari and Baskeball benchmark (in next subsection), since the expert policies are not available, it is challenging to compute DTW, ED, or KL as we did in Table 1. We evaluate the style calibration of the policy by comparing the accuracy of the style of the agent's actual trajectory with the given style. A higher value indicates better diversity of the policy. There are also other candidate metrics to evaluate state diversity or action diversity (Belkhale et al., 2024).

In the Alien and MsPacman game, we employ two styles: the area style and the range style. The former divides the map into four distinct areas, distinguishing the agent's preferences for moving toward each area. The latter models the agent's displacement trajectory on the map as a Gaussian distribution, differentiating the variance of the distribution. A higher variance indicates a tendency for the agent to move across areas, while a lower variance indicates a preference for movement within a single area. In the SpaceInvader game, we also utilize two styles: the firing rate style and the area style. For further details, please refer to Appendix B.4.

The results in Figure 2 indicate a high relevance between the agent's state-action distribution and the style. Moreover, as the relevance increases, the converged MI values also increase. The MsPacmanAR style divides the map into four areas: top-left, top-right, bottom-left, and bottom-right. The agent's trajectory exhibits clear distinctions under this style, which is reflected in the larger converged MI values, while the MsPacmanRG style represents the style based on the range of movement, which is less distinct compared to the movement area style.

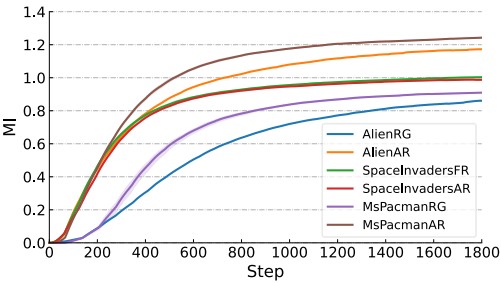

Figure 2: MI between state-action pairs and styles. FR: Fire Rate style; AR: Movement Area style; RG: Movement Range style.

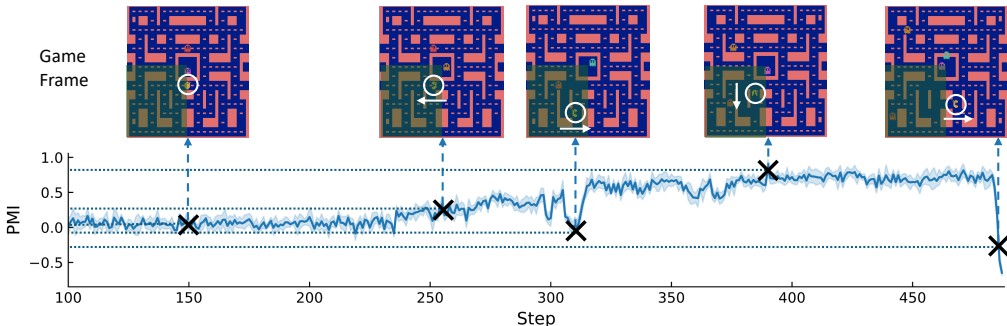

Figure 3: The PMI weight values related to style for each frame along a trajectory, with the corresponding style being the bottom-left area style (green color) in the game frame. The agent in the game frame is indicated by a white circle, and the corresponding actions are indicated by white arrows. No arrow indicates that the action is NOOP.

As shown in Figure 3, in the first frame, the agent is located at the edge of the area and does not take any action, resulting in a near-zero relevance with the style. In the second and fourth frames, the agent moves to the corresponding area and exhibits a tendency to continue moving towards that area, leading to a higher relevance with the style and higher PMI values. Conversely, in the third and fifth frames, the agent is positioned at the edge or outside of the style area and shows a tendency to move away from the style area, resulting in a lower relevance with the style.

To verify whether different styles of policies affect the cumulative reward, we conducted experiments in three different Atari environments, as shown in Table 2. The results indicate that most styles, such as those related to position and range of movement, do not impact the cumulative reward of the agent. However, some styles that are highly correlated with scoring can significantly influence the cumulative reward. For instance, in Space Invaders, the Fire style, characterized by a higher firing frequency, significantly increases the agent's cumulative reward, while a lower firing frequency reduces the cumulative reward.

Finally, we compared the calibration of several different style policies across these Atari environments, as shown in Table 3. In styles related to position and firing, BC-PMI outperformed all baselines. In styles related to range, BC-PMI achieved performance comparable to CTVAE. The reason is that the mutual information distinguishing capability of styles related to position and firing is higher (refer to Figure 2). Consequently, the trained MINE network can assign more accurate PMI values to state-action pairs, resulting in higher accuracy for the corresponding styles.

Table 2: Comparison of the cumulative reward across different styles in Atari. BC-PMI-x means the reward of the x-th style.

| Env | Style | BC | BC-PMI-1 | BC-PMI-2 | BC-PMI-3 | BC-PMI-4 |
|-----|-------|-----|----------|----------|----------|----------|
| Alien | Pos | 460.5±173.4 | 416.0±160.8 | 432.0±174.7 | 500.3±211.7 | 496.1±191.5 |
| | Range | 460.5±173.4 | 440.0±217.0 | 501.2±247.9 | 423.2±240.9 | - |
| Ms Pacman | Pos | 701.5±182.4 | 665.4±216.3 | 803.5±211.2 | 674.3±146.5 | 666.2±172.2 |
| | Range | 701.5±182.4 | 636.1±265.1 | 759.5±280.6 | 738.3±260.8 | - |
| Space Invaders | Pos | 208.2±73.0 | 178.9±54.5 | 187.0±54.9 | 200.2±56.4 | - |
| | Fire | 208.2±73.0 | 94.8±47.2 | 207.9±70.3 | 270.6±86.7 | - |

Table 3: Comparison of style calibration across different styles in Atari.

| Env | Style | BC | CBC | CTVAE | InfoGAIL | SORL | BC-PMI |
|-----|-------|-----|-----|-------|----------|------|--------|
| Alien | Pos | 0.23±0.11 | 0.43±0.04 | 0.52±0.04 | 0.34±0.11 | 0.31±0.07 | **0.61±0.08** |
| | Range | 0.31±0.09 | 0.51±0.13 | **0.57±0.09** | 0.31±0.11 | 0.27±0.07 | 0.55±0.12 |
| MS Pacman | Pos | 0.17±0.05 | 0.48±0.07 | 0.55±0.04 | 0.33±0.05 | 0.27±0.08 | **0.57±0.07** |
| | Range | 0.33±0.09 | 0.53±0.11 | **0.61±0.05** | 0.29±0.09 | 0.30±0.12 | 0.56±0.14 |
| Space Invaders | Pos | 0.36±0.06 | 0.71±0.08 | 0.66±0.10 | 0.41±0.06 | 0.37±0.04 | **0.83±0.06** |
| | Fire | 0.27±0.07 | 0.41±0.08 | 0.47±0.06 | 0.36±0.08 | 0.34±0.09 | **0.51±0.10** |

## 5.4 PROFESSIONAL BASKETBALL PLAYER DATASET

In this experiment, we validate our method on the dataset of a collection of professional basketball player trajectories (Zhan et al., 2020) with the goal of recovering policies that can generate trajectories with diverse player-movement styles. The basketball trajectories are collected from tracking real players in the NBA. We primarily focus on two movement styles: (1) The **Destination**, which is the distance from the final position to a fixed destination on the court (e.g. the basket), and (2) The **Curvature**, which measures the players propensity to change directions.

Figure 4 and Figure 5 present the calibration results of different styles for the BC-PMI algorithm in the dimensions of *Destination* and *Curvature*. The environment is a half-court basketball setting, where each player's trajectory can be categorized into different styles based on their movement destination and curvature. In Figure 4, Destination 1 is close to the ball frame, Destination 2 is in the middle count, and Destination 3 is far from the ball frame. The three destinations are separated by green lines in the figure. Similarly, three different movement curvature styles are illustrated in Figure 5. The results indicate that the BC-PMI algorithm can effectively imitate policies of different movement styles from real human data.

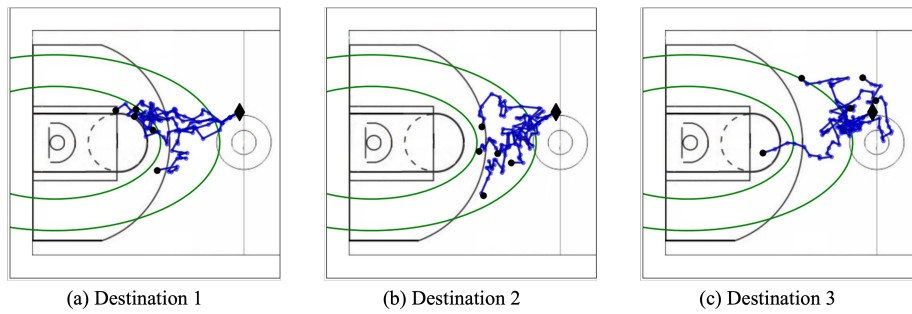

(a) Destination 1         (b) Destination 2         (c) Destination 3

Figure 4: Visualization of different destination styles.

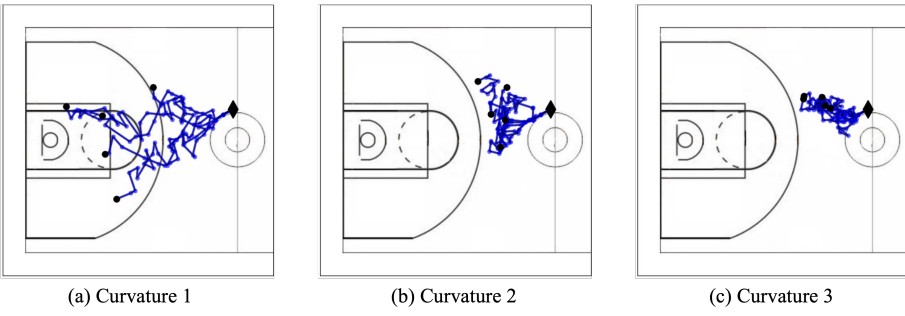

(a) Curvature 1        (b) Curvature 2        (c) Curvature 3

Figure 5: Visualization of different curvature styles.

Table 4: Comparison of style calibration (%) across different styles in Basketball

| Style | Class | BC | CBC | CTVAE | InfoGAIL | SORL | BC-PMI |
|---|---|---|---|---|---|---|---|
| Destination | Class 1 | 31.1 | 81.6 | 78.3 | 79.6 | 80.9 | **93.3** |
| | Class 2 | 42.7 | 84.4 | 81.2 | 75.5 | 76.1 | **92.7** |
| | Class 3 | 19.4 | 73.1 | 77.8 | 74.5 | 79.5 | **89.8** |
| Curvature | Class 1 | 17.1 | 67.3 | 58.4 | 60.9 | 51.6 | **79.4** |
| | Class 2 | 30.6 | 66.2 | 58.7 | 57.2 | 49.4 | **80.9** |
| | Class 3 | 41.8 | 71.6 | 62.1 | 62.7 | 53.8 | **81.6** |

We compare style calibration for 3 classes of *Destination* and 3 classes of *Curvature* in Table 4. Due to the lack of style label information, BC can only learn an approximately average strategy, so it should be evenly distributed among various styles. However, due to factors such as initialization position, the samples in each category do not strictly follow a uniform distribution. Among the various methods with diversity mechanisms, the BC-PMI method, which focuses on state-action pairs highly relevant to the style, surpasses other baselines in terms of style calibration.

## 6 DISCUSSION

### 6.1 CONCLUSION

In this paper, we investigate how to recover diverse policies from a set of expert trajectories. We propose a new diverse policy recovering method by leveraging the relevance of the state-action pair with the trajectory styles. The highlight of our method lies in our approach to the problem of policy diversity from a different perspective, which involves the introduction of Pointwise Mutual Information to model the relevance between each $(s, a)$ pair and the style. By utilizing a unique and straightforward approach, we achieved results that surpassed previous state-of-the-art methods.

### 6.2 LIMITATIONS AND FUTURE WORK

In our setting, our goal is to recover policies from diverse offline data, assuming that the data within the trajectories already meet the performance requirements. If the policy that generates the offline trajectories performs poorly, our method cannot further enhance the performance of the learned policy. Hence, we propose exploring the combination of PMI and RL in future research. We believe that integrating PMI weights with RL methods can enhance diversity while simultaneously improving performance. Numerous studies have already demonstrated the effectiveness of promoting diversity in RL (Hong et al., 2018; Eysenbach et al., 2018; Parker-Holder et al., 2020; Peng et al., 2020; Tang et al.; Yao et al., 2024). Another potential direction is to incorporate disentanglement methods (Cao et al., 2022; Locatello et al., 2019) between the stylized code and the state-action pair. For example, we can adapt $R_{ij}$ and MED to weight the BC loss by replacing MI with PMI.

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

# A    THEORETICAL DERIVATION

## A.1    PROOF OF PROPOSITION 1

*Proof.* (a) We first introduce the Gibbs' inequality, which was proved in (Brémaud, 2012).

**Lemma 1** (Gibbs' inequality). *Let $p(x)$, $q(x)$ be the probability mass function, fixed $p(x)$, that the object function $-\mathbb{E}_{x \sim p(x)} \log q(x) = \sum_x -p(x) \log q(x)$ takes the minimum when $q(x) = p(x)$.*

By Lemma 1, we know

$$\mathcal{L}_{BC}(\theta) = \mathbb{E}_{(s,a) \sim \mathcal{D}_e} [-\log \pi_\theta(a|s)] = \mathbb{E}_{s \sim \mathcal{D}_e} \left[ \mathbb{E}_{a|s \sim p(a|s)} [-\log \pi_\theta(a|s)] \right], \tag{16}$$

which takes the minimum when $\pi_\theta(a|s) = p(a|s)$.

When $I(Z; S, A) = 0$, it means that the latent variable $Z$ is independent of the state-action pair $(S, A)$, and the PMI term in the BC-PMI objective becomes:

$$\frac{p(z|s,a)}{p(z)} = \frac{p(z)}{p(z)} = 1. \tag{17}$$

Hence,

$$\begin{aligned}
\mathcal{L}_{\mathrm{BC-PMI}}(\theta) &= \mathbb{E}_{(s,a,z) \sim \mathcal{D}_e} [-\log \pi_\theta(a|s,z)] \\
&= \mathbb{E}_{s \sim \mathcal{D}_e} \left[ \sum_a \sum_z -p(a|s)p(z|a,s) \log \pi_\theta(a|s,z) \right] \\
&= \mathbb{E}_{s \sim \mathcal{D}_e} \left[ \sum_z \sum_a -p(a|s)p(z) \log \pi_\theta(a|s,z) \right] \\
&= \mathbb{E}_{s \sim \mathcal{D}_e} \left[ \sum_z \left[ p(z)(\sum_a -p(a|s) \log \pi_\theta(a|s,z)) \right] \right],
\end{aligned} \tag{18}$$

where the third equation is because $z$ is independent with $(s, a)$. Since $p(z)$ is non-negative, when $\sum_a -p(a|s) \log \pi_\theta(a|s,z)$ takes the minimum for all $z$, the objective reach its the minimum. Using Lemma 1 again, we can see when $\pi_\theta(a|s,z) = p(a|s)$, $\mathcal{L}_{\mathrm{BC-PMI}}(\theta)$ reach its minimum.

(b) when $H(Z|S, A) = 0$, it means that given a state-action pair $(s, a)$, the style variable $Z$ can be determined with high certainty. In other words, for any $(s, a, z) \sim \mathcal{D}_e$, we have:

$$p(z|s,a) = \mathbb{1}(z = z_{s,a}), \tag{19}$$

where $z_{s,a}$ is the unique style label corresponding to $(s, a)$.

Assuming there are $K$ styles in the Dataset. Let $z_\tau$ be the unique style label corresponding to trajectory $\tau$, $D_e^{(i)}$ be the subset of state-action pairs with style label $i$ and $\tau \in D_e^{(i)}$ means the whole trajectory from $D_e^{(i)}$. Denote

$$\mathcal{L}_{\mathrm{BC}}^{(i)}(\theta) = \mathbb{E}_{(s',a') \sim D_e^{(i)}} [-\log \pi_\theta(a'|s',i)] = \frac{1}{|D_e^{(i)}|} \sum_{\tau \in D_e^{(i)}} \sum_{(s',a') \in \tau} -\log \pi_\theta(a'|s',i). \tag{20}$$

We have

$$
\begin{aligned}
\mathcal{L}_{\mathrm{BC-PMI}}(\theta) &= \underset{(s,a,z)\sim\mathcal{D}_e}{\mathbb{E}} \left[ -\log \pi_\theta(a|s,z) \cdot \frac{p(z|s,a)}{p(z)} \right] \\
&= \underset{(s,a,z)\sim D_e}{\mathbb{E}} \left[ -\log \pi_\theta(a|s,z) \frac{\mathbb{1}(z=z_{s,a})}{p(z)} \right] \\
&= \frac{1}{|D_e|} \sum_{\tau\in D_e} \sum_{i=1}^{K} \mathbb{1}(z_\tau=i) \sum_{(s',a')\in\tau} \left[ -\log \pi_\theta(a'|s',i) \cdot \frac{1}{p(z=i)} \right] \\
&= \frac{1}{|D_e|} \sum_{i=1}^{K} \sum_{\tau\in D_e} \sum_{(s',a')\in\tau} \mathbb{1}(z_\tau=i) \left[ -\log \pi_\theta(a'|s',i) \cdot \frac{|D_e|}{|D_e^{(i)}|} \right] \\
&= \sum_{i=1}^{K} \sum_{\tau\in D_e^{(i)}} \frac{1}{|D_e^{(i)}|} \sum_{(s',a')\in\tau} -\log \pi_\theta(a'|s',i) \\
&= \sum_{i=1}^{K} \mathcal{L}_{\mathrm{BC}}^{(i)}(\theta)
\end{aligned}
\tag{21}
$$

Note that by the definition, $\mathcal{L}_{\mathrm{BC}}^{(i)}(\theta)$ is the behavior cloning loss on the subset of data with style label $i$. Hence, the last equation above means when $H(Z|A,S)$ equals 0, the BC-PMI objection function can be viewed as separating the $(s,a)$ by the style $z$ and optimizing it respectively. This is equivalent to the clustering-based behavior cloning objective, where each trajectory is first assigned to a specific style cluster based on its style label, and then behavior cloning is performed within each cluster. This result reveals the close connection between BC-PMI and clustering-based behavior cloning in the extreme case of zero style overlap. □

## B   IMPLEMENTATION DETAILS

### B.1   COMPUTATIONAL RESOURCE

All experiments in this paper are implemented with PyTorch and executed on NVIDIA Tesla T4 GPUs. All the runs in experiments use 5 random seeds.

### B.2   COMMON HYPERPARAMETERS

Table 5: Common hyperparameters setting.

| Hyperparameter | Circle 2D | MsPacman | SpaceInvaders | Basketball |
|---|---|---|---|---|
| learning rate | 0.001 | 0.001 | 0.001 | 0.0002 |
| optimizer | Adam | Adam | Adam | Adam |
| epoch | 10 | 30 | 30 | 30 |
| batch size | 128 | 512 | 512 | 128 |
| hidden dim | 32 | 64 | 64 | 128 |

### B.3   MINE NETWORK STRUCTURE

Figure 6 shows the MINE network structure, which is used in Atari tasks.

### B.4   STYLES IN SPACEINVADERS GAME

In the SpaceInvaders game, we employ two styles: the area style and the fire rate style. The former divides the map into three distinct areas, distinguishing the agents preferences for moving toward each area. The latter divides the agent's fire rate into three levels: (1) [0, 0.1); (2) [0.1, 0.3) and (3) [0.3, 1].

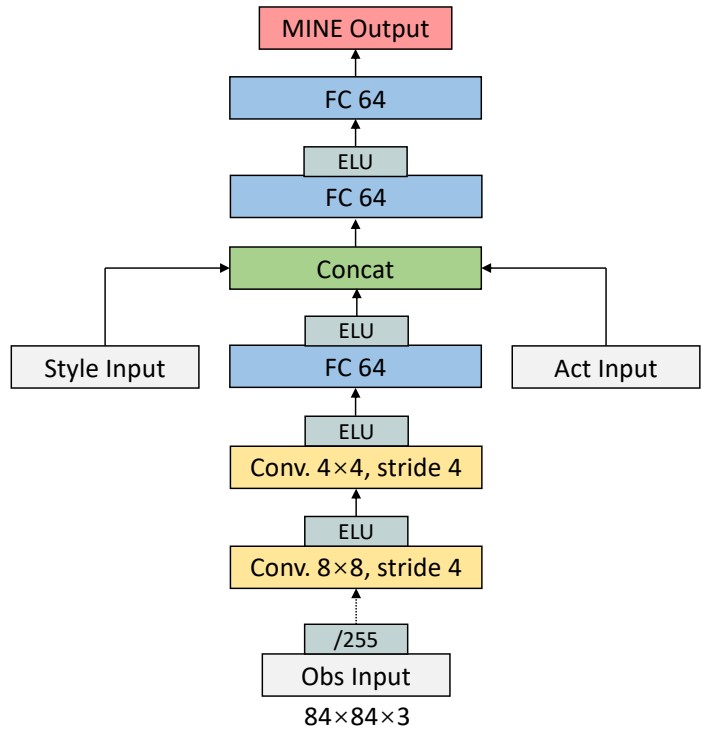

Figure 6: Architecture of the MINE networks used in the Atari environment.

## C Comparison of Weights Between the BC-PMI and the CBC Method

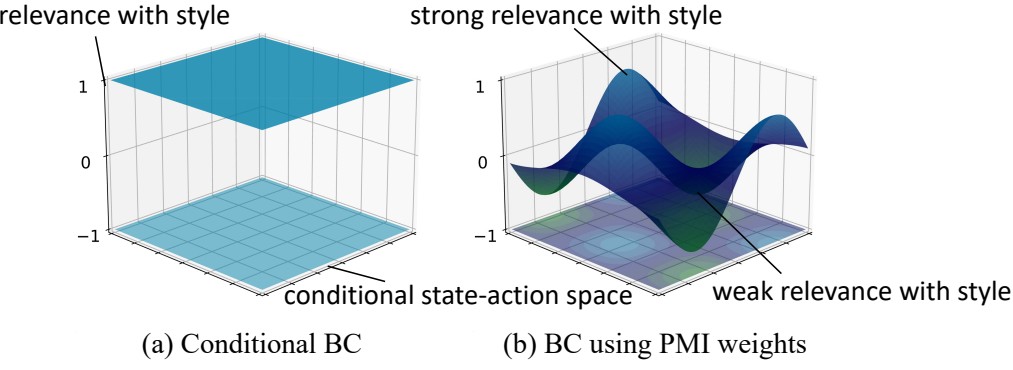

(a) Conditional BC          (b) BC using PMI weights

Figure 7: The weight distributions for $(s, a)$ pairs in different imitation learning paradigms are given as follows: (a) In clustering or condition based methods, the weights for $(s, a)$ pairs belonging to the same style are set to 1, while the weights for other pairs are set to 0; (b) In the proposed PMI weighting method, the weight distribution is approximated using Pointwise Mutual Information, which considers the relevance between each $(s, a)$ pair and the style.

Lacking consideration to recognize the relevance of each state-action pair to the style is quite common in various methods (e.g., vanilla BC in Eq.(1)). The unsupervised clustering-based or simply conditional methods can be regarded as using indicator functions to differentiate state-action pairs and assign corresponding style labels (Figure 7a). State-action pairs that belong to a specific style weight 1, while pairs not belonging to that style weight 0, which can be denoted as the following

objective:

$$\min_{\pi} \mathbb{E}_{(s,a)\sim\mathcal{D}_e} \big[ -\log\pi(a|s)\cdot\mathbb{1}_z\{\lambda(s,a)\}\big], \tag{22}$$

where $\lambda(s,a)$ is a clustering function. To leverage the diversity information, our proposed method utilizes PMI to approximate the relevance between $(s,a)$ pairs and styles, thereby obtaining more appropriate weights (Figure 7b).

# D  THE LABEL FUNCTION

In our method, we focus on behavioral styles that can be represented by a trajectory-level label function, denoted as $\mathcal{J}$, which is used to determine the style to which a given trajectory belongs:

$$\mathcal{J}(\tau_e) = \mathbb{1}\Big\{\Big\|\sum\nolimits_{(s,a)\sim\tau_e}\lambda(s,a)\Big\|_2 > c\Big\}, \tag{23}$$

where $\lambda$ is a state-action-level label function used to distinguish the behavioral styles, and $c$ is a style threshold. In practice, we may only use a segment of the trajectory $\mathcal{J}(\tau_e^{t:t+i})$ to get the labels.

In this paper, following the common routine, we approach this by specifying a task-relevant style embedding (represented by $z$) in advance. For example, in atari, similar to Wu et al. (2023), we pre-define the styles such as fire rate; in basketball, we define the "destination" and "curvature," which are commonly used as the style labels or metrics (Zhan et al., 2020; Li et al., 2021). A Python-style label function for fire rate is presented as follows:

```python
for traj in range(len(all_trajs)):
    traj_name = "{}.txt".format(all_trajs[traj])
    with open(os.path.join(data_dir, traj_name), 'r') as f1, open(os.path
                                        .join(label_dir, traj_name), 'w'
                                        ) as f2:
        lines = f1.read().splitlines()
        idx = np.arange(len(lines))
        sub_idx = [idx[i: i+sublen] for i in range(0, len(idx)-sublen,
                                            sublen)]
        max_line = len(idx)-sublen
        for si in sub_idx:
            n_fire = 0
            fire_rate = None
            count = 0
            for sj in si:
                data = lines[sj+1].split(",")[:8]
                data = [s.strip() for s in data]
                if len(data) >= 8 and data[-3] != 'null':
                    count += 1
                    action = int(data[-3])
                if action == 1:
                    n_fire += 1
            fire_rate = round(float(n_fire / count), 2)
            if fire_rate < 0.1:
                style = 1
            elif fire_rate >= 0.1 and fire_rate <= 0.3:
                style = 2
            elif fire_rate > 0.3:
                style = 3

            for sj in si:
                data = lines[sj+1].split(",")[:8]
                data = [s.strip() for s in data]
                data.insert(6, str(style))

                lines[sj+1] = ','.join(data)

        f2.write('\n'.join(lines[:max_line]))
```

A significant benefit is that labeling functions are often simple scripts that can be quickly applied to the dataset, which is much cheaper than manual annotations and more reliable than unsupervised methods. In addition, we conducted a data analysis experiment on the label function, comparing the classification ratios of each category label across the Atari dataset under each label function. The results in Table 6 show that the data differences for each category label are not significant, which is beneficial for the accuracy of the MINE network estimation. If the data for a particular category label is too sparse, the estimated mutual information value may have a considerable error.

Table 6: The proportion of various styles of data labeled using the label function.

| Env | Style | Category 1 | Category 2 | Category 3 | Category 4 |
|---|---|---|---|---|---|
| Alien | Pos | 15.19% | 34.67% | 36.82% | 13.32% |
| | Range | 37.08% | 43.82% | 19.10% | - |
| Ms Pacman | Pos | 15.37% | 33.76% | 31.47% | 19.40% |
| | Range | 26.97% | 43.82% | 29.21% | - |
| Space Invaders | Pos | 32.35% | 32.35% | 35.29% | - |
| | Fire | 18.52% | 37.04% | 44.44% | - |

Our method focuses on the relevance between each $(s, a)$ pair in a trajectory and the style. When the MINE function is well-fitted, samples that are completely irrelevant to the style will have PMI values close to zero, implying that these samples have minimal impact on the policy. However, noise introduced by the label function may affect the training of the MINE function. To analyze the impact of noise sample ratios on calibration, we use the *position* style in Space Invasers environment as an example. We define samples in a trajectory that fall outside the position range as noise samples. Table 7 presents the proportion of noise samples for different styles and the corresponding calibration values. The results indicate that as the noise ratio increases, the improvement of BC-PMI over CBC becomes more significant.

Table 7: Comparison of calibration Under different noise ratios.

| Pos | Label Noise | CBC | BC-PMI |
|---|---|---|---|
| Left | 16.25% | 0.75 | 0.84 |
| Middle | 31.23% | 0.62 | 0.80 |
| Right | 19.28% | 0.74 | 0.86 |

# E    ADDITIONAL RESULTS

## E.1    THE NUMBER OF STYLES

We increased the fire rate to 6 categories in the Space Invaders environment to observe the impact of the increase in style categories on calibration. The results are shown in Table 8. Increasing the number of style categories may slightly decrease calibration, but compared to CBC, there is still a noticeable improvement.

## E.2    IMPLEMENTATION OF OTHER VARIANT METHODS

We implemented a simple version of the GAIL algorithm combined with PMI. Specifically, we first developed a Conditional GAIL (Ho & Ermon, 2016) (CGAIL) variant, where joint information $(s, a, z)$ is used to determine whether a sample was generated by the actor. During interactions between the actor and the environment, the style label $z$ is also provided as input when selecting actions. The PMI value is incorporated as a weight for the reward generated by the discriminator.

Table 8: Comparison of calibration for more style categories.

| Fire Rate | Proportion | CBC | BC-PMI |
|---|---|---|---|
| [0.0, 0.1) | 18.52% | 0.39±0.07 | 0.45±0.07 |
| [0.1, 0.2) | 22.17% | 0.41±0.09 | 0.52±0.05 |
| [0.2, 0.3) | 14.85% | 0.41±0.11 | 0.46±0.07 |
| [0.3, 0.4) | 17.42% | 0.31±0.04 | 0.43±0.08 |
| [0.4, 0.5) | 15.89% | 0.36±0.05 | 0.39±0.09 |
| [0.5, 1.0] | 11.13% | 0.46±0.07 | 0.53±0.05 |

When the reward is positive, samples with a higher relevance between $(s, a)$ and $z$ receive a greater reward. The framework of the proposed method is illustrated in Figure 8.

For the DWBC (Xu et al., 2022) method, the original DWBC algorithm uses $d(s, a, \log \pi)$ to determine whether the current sample is generated by the expert policy. In our setting, we use $d(s, a, z)$ to determine whether the current sample belongs to style $z$. We compare the influence of the weights generated by the discriminator and the PMI weights on policy diversity. Finally, the BC-Classifier method uses the softmax output of a classifier to estimate $p(z|s, a)$ as the weight of the sample. The results are shown in Table 9.

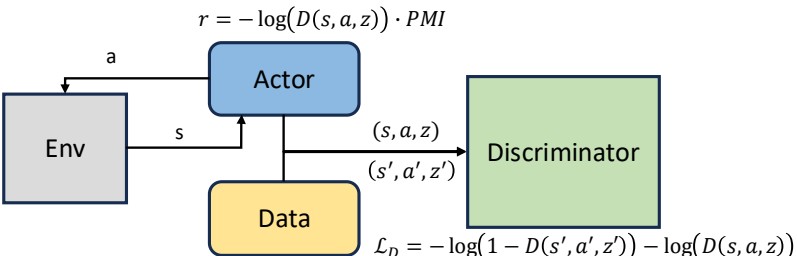

Figure 8: The Preliminary Framework of GAIL-PMI Method.

Table 9: Comparison of style calibration across different variant methods in Atari

| Env | Style | CGAIL | CGAIL-PMI | BC-Classifier | DWBC | BC-PMI |
|---|---|---|---|---|---|---|
| Alien | Pos | 0.31±0.09 | 0.39±0.13 | 0.47±0.07 | 0.44±0.05 | **0.61±0.08** |
| | Range | 0.37±0.12 | 0.37±0.10 | **0.55±0.10** | 0.41±0.08 | **0.55±0.12** |
| MS Pacman | Pos | 0.39±0.11 | 0.38±0.08 | 0.54±0.09 | 0.40±0.09 | **0.57±0.07** |
| | Range | 0.41±0.09 | 0.45±0.13 | **0.57±0.11** | 0.39±0.10 | 0.56±0.14 |
| Space Invaders | Pos | 0.48±0.12 | 0.60±0.11 | 0.75±0.08 | 0.64±0.11 | **0.83±0.06** |
| | Fire | 0.44±0.07 | 0.49±0.11 | 0.40±0.07 | 0.33±0.09 | **0.51±0.10** |

