# OpenReview forum: "Diverse Policies Recovering via Pointwise Mutual Information Weighted Imitation Learning"
_ICLR.cc/2025/Conference — ICLR 2025 Poster_

### Official Review · Reviewer_BUhV · 2024-11-02

**Soundness:** 3
**Presentation:** 3
**Contribution:** 4
**Rating:** 8
**Confidence:** 4

**Summary:**

This paper investigates how to recover diverse policies from expert trajectories, proposing a new method that leverages the relevance of state-action pairs to trajectory styles. By introducing Pointwise Mutual Information to model this relationship, the method approaches the problem of policy diversity from a different perspective, ultimately achieving results that surpass previous state-of-the-art methods.

**Strengths:**

1. The motivation is simple but effective, and it will be highly beneficial for the efficient use of imitation learning data in subsequent robot tasks, contributing significantly to the development of the community.
2. The paper is well-written and easy to follow.
3. The experimental results are extensive and convincing.

**Weaknesses:**

1. I would like to know if there are other reweighting methods that can be used for comparison. If so, could some comparative experiments be included?
2. Additionally, the current method evaluates the diversity of state-action pairs. As far as I understand, there are also studies specifically focused on state diversity and action diversity. I wonder if these could serve as a standard for comparison?

[1] Suneel Belkhale, Yuchen Cui, Dorsa Sadigh. Data quality in imitation learning. NeurIPS 2023

**Questions:**

Please refer to the weakness.

---

> ### Author Response · Authors · 2024-11-21
>
> We thank the reviewer for the insightful and valuable feedback. We explain the concerns point by point below.
>
> **W1: I  would like to know if there are other reweighting methods that can be used for comparison. If so, could some comparative experiments be included?**
>
> R1: In fact, there are some weighting methods can be compared, inluding: DWBC[1] weights the IL loss using the discriminator, and [2] weights via the old policy).
> Due to time limited, we only implement DWBC as our competitor. For the DWBC method, the original DWBC algorithm uses $d(s, a, log \pi)$ to determine whether the current sample is generated by the expert policy. In our setting, we use $d(s, a, z)$ to determine
> whether the current sample belongs to style z. The experiment results is shown in Table 9. We can see that DWBC performes better than BC but worse than BC-PMI. We have add the results in Appendix E.
>
> [1] Xu, Haoran, et al. "Discriminator-weighted offline imitation learning from suboptimal demonstrations." International Conference on Machine Learning. PMLR, 2022.
>
> [2] Sasaki, Fumihiro, and Ryota Yamashina. "Behavioral cloning from noisy demonstrations." International Conference on Learning Representations. 2020.
>
> **W2: Additionally, the current method evaluates the diversity of state-action pairs. As far as I understand, there are also studies specifically focused on state diversity and action diversity. I wonder if these could serve as a standard for comparison?**
>
> R2: We have incorporated some state diversity (e.g., area style in Atari) and action diversity (e.g., fire rate style in Atari) in our experiment. We believe this reflects the diversity evaluated under state-only or action-only perspectives to some extent. Introducing a more theoretical definition of state or action diversity into the evaluation is indeed an interesting idea [1]. We have further discussed other possible metrics for state diversity or action diversity in the revised version.
>
> [1] Suneel Belkhale, Yuchen Cui, Dorsa Sadigh. Data quality in imitation learning. NeurIPS 2023

---

> > ### Comment · Reviewer_BUhV · 2024-11-28
> > **Official Comment by Reviewer BUhV**
> >
> > Thanks for your response!
> >
> > I look forward to seeing more discussion on other possible metrics for state diversity or action diversity in the revised version.
> >
> > Another promising way is incorporating the current method with further embodiment tasks[1-3], which is the primary reason for this paper's value from my point of view. I hope the author can extend the proposed method into the VLA fine-tuning task in the future, which is an emergent demand for Embodied AI.
> >
> > [1] ReMix: Optimizing Data Mixtures for Large Scale Imitation Learning
> > [2] Data Scaling Laws in Imitation Learning for Robotic Manipulation
> > [3] Heterogeneous Pre-trained Transformer (HPT) as Scalable Policy Learner.

---

> > > ### Author Response · Authors · 2024-11-28
> > > **Official Comment by Authors**
> > >
> > > Thank you very much for your thoughtful feedback. Your insights are invaluable in enhancing the quality of our work.
> > > We will further discuss other possible metrics for state or action diversity in the paper.
> > > Regarding incorporating BC-PMI into embodiment tasks, we appreciate your suggestion and find it intriguing.
> > > We will discuss this potential direction in the paper and try to extend BC-PMI to the VLA fine-tuning task in future research.

---

> ### Author Response · Authors · 2024-11-25
>
> Dear Reviewer BUhV,
>
> Thanks again for your valuable review. We are wondering whether our rebuttal has addressed your concerns. We would love an active discussion with you and hope to address any remaining concern or question you may have.
>
> Best Regards, Authors

---

### Official Review · Reviewer_fKFr · 2024-11-02

**Soundness:** 3
**Presentation:** 3
**Contribution:** 3
**Rating:** 8
**Confidence:** 4

**Summary:**

The paper introduces Behavioral Cloning with Pointwise Mutual Information Weighting (BC-PMI), a new approach for recovering diverse policies in imitation learning. Here, the expert data consists of trajectories coming from different experts, each reflecting a unique approach or style within the task. The authors’ key idea is that, even within a single style, not all state-action pairs equally represent that style’s characteristics. To address this, BC-PMI weights state-action pairs based on their relevance to the style, calculated using pointwise mutual information between $(s,a)$ pairs and the style code $z$. These weighted pairs are then used to learn a style-conditioned policy $\pi(a|s,z)$ using behavioral cloning. This selective weighting allows the model to prioritize pairs that are most representative of the target behavior, thereby enhancing its ability to learn style-conditioned policies that more accurately capture the style within expert trajectories.

The paper also provides some theoretical insights showing that BC-PMI smoothly interpolates between traditional behavioral cloning and clustering-based behavior cloning, depending on the mutual information between style and state-action pairs.

The authors evaluate BC-PMI in three settings: Circle 2D, a simple 2D setup; Atari Games (Alien, MsPacman, and Space Invaders); and the Professional Basketball Dataset, featuring NBA player movement data. Results show that BC-PMI achieves high style calibration across various metrics, outperforming baselines. However, it is worth noting that the baselines operate in an unsupervised manner and do not have access to the style labels available to BC-PMI.

**Strengths:**

1. The paper is well written and clearly communicates its key ideas.
2. The idea of selectively weighting state-action pairs based on their relevance to a given style, rather than treating entire trajectories uniformly, is both intuitive and impactful. Existing methods typically operate at the trajectory level, but this paper demonstrates that not all state-action pairs within a trajectory equally represent the intended style.

**Weaknesses:**

1. A key disadvantage of the method is that it requires style markers for each trajectory in the training dataset. These can be expensive to obtain. A brief note is made about using labeling functions to obtain these markers, but no experiments are included. This also makes the comparison with existing works unfair since they are unsupervised methods. To make the comparisons fair, the authors should include the supervision in other methods, e.g., they can train the posterior network in InfoGAIL using the style labels.

2. The experimental section is relatively brief, lacking analysis on the method’s performance as the number of styles increases or when labels are noisy. The latter case is quite important given that labels generated by programmable labeling functions (as mentioned in L296) may be imperfect. For a comprehensive understanding of these factors, it would be beneficial to include experiments that measure calibration as the number of styles increases, as well as tests assessing robustness to label noise introduced by such labeling functions.

**Questions:**

1. Why is MINE used to estimate the pointwise mutual information? This can be estimated by simply learning a classifier across different styles, since you are not interested in the $I(S,A;Z)$ rather $p(z|s,a)$. It would be valuable to compare the calibration performance of BC-PMI using a classifier-based approach against the current method with MINE’s statistic network, to assess if simpler alternatives might achieve similar results.
2. What are the calibration metrics used in Tables 3/4? Is it DTW, or ED, or KL?

---

> ### Author Response · Authors · 2024-11-21
>
> We thank the reviewer for the insightful and valuable feedback. We explain the concerns point by point below.
>
> **W1.1: A key disadvantage of the method is that it requires style markers for each trajectory in the training dataset. These can be expensive to obtain. A brief note is made about using labeling functions to obtain these markers, but no experiments are included.**
>
> R1.1: In our experiments, we utilized a labeling function, which takes a minimal amount of time to label the trajectory since it only runs once for each data point. Following common practice, we approached this by specifying a task-relevant style embedding (represented by 𝑧) in advance. In Atari, similar to Wu et al. (2023), we pre-defined styles such as fire rate. In basketball, we defined "destination" and "curvature" as the style labels, consistent with the styles defined in Zhan et al. (2020).
> We have made this clearer in Appendix D and provided a Python-style labeling function for fire rate to better illustrate this process.
>
> **W1.2: This also makes the comparison with existing works unfair since they are unsupervised methods. To make the comparisons fair, the authors should include the supervision in other methods, e.g., they can train the posterior network in InfoGAIL using the style labels.**
>
> R1.2: Actually in the our implement of InfoGAIL and SORL, in order to make the comparison more fair, we concatenate the state and style as input to the network. We made this more clearer in the revision of the version (as highligted in Sec 5.1).
>
> [1] Wu, Shuang, et al. "Quality-similar diversity via population based reinforcement learning." The Eleventh International Conference on Learning Representations. 2023.
>
> [2] Zhan, Eric, et al. "Learning calibratable policies using programmatic style-consistency." International Conference on Machine Learning. PMLR, 2020.
>
> **W2: The experimental section is relatively brief, lacking analysis on the method’s performance as the number of styles increases or when labels are noisy. The latter case is quite important given that labels generated by programmable labeling functions (as mentioned in L296) may be imperfect. For a comprehensive understanding of these factors, it would be beneficial to include experiments that measure calibration as the number of styles increases, as well as tests assessing robustness to label noise introduced by such labeling functions.**
>
> A2: We have conducted experiments and reported the results in Table 7 and Table 8. In Table 7, we compare the calibration for styles with different noise to analyze the impact
> of noisy sample ratios. In Table 8, we increased the number of styles for the fire rate in Atari and compared the performance of CBC and BC-PMI. The experiments show that in both cases, BC-PMI performs better than CBC. This is expected since PMI weights the imitation learning loss according to mutual information, which implicitly learns the distribution between styles (with noise) and state-action pairs.

---

> > ### Author Response · Authors · 2024-11-21
> >
> > **Q1: Why is MINE used to estimate the pointwise mutual information? This can be estimated by simply learning a classifier across different styles, since you are not interested in the $I(S,A,Z)$ rather $p(z|s,a)$ . It would be valuable to compare the calibration performance of BC-PMI using a classifier-based approach against the current method with MINE’s statistic network, to assess if simpler alternatives might achieve similar results.**
> >
> > A1: The weight we propose in the paper is $\log \frac{p(z|s,a)}{p(z)}$. We acknowledge an alternative way to estimate this is by training a classifier to estimate $p(z|s,a)$ and then calculating $\log \frac{p(z|s,a)}{p(z)}$ during training (the prior $p(z)$ is available). We have conducted this experiment, and the results are provided in Table 9 in Appendix E.2. Our findings indicate that this alternative method is comparable to CBC but slightly weaker than our proposed method. Experimentally, we conclude that directly estimating $\log \frac{p(z|s,a)}{p(z)}$ may be more appropriate than estimating $p(z|s,a)$ and then calculating $\log \frac{p(z|s,a)}{p(z)}$. We have added this discussion to the revised version and thank you again for the suggestion!
> >
> > **Q2: What are the calibration metrics used in Tables 3/4? Is it DTW, or ED, or KL?**
> >
> > A2: In Atari and Baskeball benchmark, since the expert policies are not available, it is challenging to compute DTW, ED, or KL as we did in Table 1.
> > We evaluate the style calibration of the policy by comparing the accuracy of the style of the agent's actual trajectory with the given style.
> > A higher value indicates better diversity of the policy.
> >  Since "expert policies" in the real world are not available and unique (as in Table 1), it is challenging to compute DTW, ED, or KL as we did in Table 1.
> >
> > We apologize for any misunderstanding this may have caused and have clarified this in the revised version of the paper.

---

> ### Author Response · Authors · 2024-11-25
>
> Dear Reviewer fKFr,
>
> Thanks again for your valuable review. We are wondering whether our rebuttal has addressed your concerns. We would love an active discussion with you and hope to address any remaining concern or question you may have.
>
> Best Regards, Authors

---

> > ### Comment · Reviewer_fKFr · 2024-11-27
> >
> > I'd like to thank the authors for the detailed rebuttal. The experiments with noisy style labels, increased styles, and more baselines strengthen the experimental analysis of the paper. I do have two comments though:
> > 1. I did not understand the explanation of the calibration metrics: "We evaluate the style calibration of the policy by comparing the accuracy of the style of the agent's actual trajectory with the given style". What does "accuracy of the style" mean?
> > 2. The section on labeling functions is weak in my opinion. It seems the functions are defined after knowledge of the styles. Is there some way to discover styles without knowing the styles beforehand?
> >
> > Despite these comments, I have decided to raise my score since most of my queries were answered.

---

> > > ### Author Response · Authors · 2024-11-27
> > >
> > > We appreciate the feedback from the reviewer. We provide explanations for the remaining two concerns below:
> > >
> > > 1. "Accuracy of the style" refers to the probability that the trajectory generated by the policy belongs to the given style after the style label is provided.
> > > We calculate the proportion of trajectories generated by the policy, given a style label, that actually belong to that style.
> > > Apologies for the misunderstanding caused by the unclear statement.
> > > We will revise this description in the paper for clarity.
> > >
> > > 2. Firstly, we use the label function to ensure that the recovered policy style is controllable.
> > > Unsupervised methods can discover styles, but such methods are often uncontrollable and unreliable.
> > > Secondly, previous work, which has largely relied on unsupervised learning, has struggled to scale to the complex styles commonly found in real-world applications, such as movement range in Atari games or motion curvature in basketball.
> > >
> > > We hope our explanations address your concerns. If you have any further questions, please feel free to continue the discussion with us.

---

### Official Review · Reviewer_ihL7 · 2024-11-04

**Soundness:** 2
**Presentation:** 3
**Contribution:** 3
**Rating:** 6
**Confidence:** 2

**Summary:**

In this work, the authors focus on developing a method for imitation learning better diversity of policy. It works under assumption that the expert trajectories in training are collected by stylized experts. The key is to using a pointwise mutual information-based weighting strategy to determine the policy importance. The state-action pair with higher posterior probability is given higher importance. The proposed method achieves good experiment results and also is demonstrated to be covered in the extreme cases, such as zero mutual information or no overlap among different policy styles in a single state-action pair. For the extreme cases, the theoretical evidence is provided to show that the proposed strategy degrades to usual strategy without failing.

**Strengths:**

1. The proposed method is motivated by an intuitive insight and the failure of existing alternatives. The use of PMI and MINE are well connected for the purpose.
2. The proposed method achieves good experimental results on the datasets of CIRCLE2D , Atari and Basketball Player dataset.

**Weaknesses:**

1. Given that stylized experts make unbalanced trajectory generation in training, it might be helpful to discuss the disentanglement between the stylized code and the state-action pair. Some previous works have built related benchmark for the purpose such as [1] and [2]. Though the benchmarks may not be originally designed for imitation learning, the corresponding metrics might be worth reference. Such evaluation can be adapted by measuring the mutual information or other metrics between the stylized code and the pair.

2. It remains not unclear, at least not generally convincing across datasets, that PMI shows a significant advance than the usual MI for BC. More evidence or discussion about this can be helpful to enhance the significance of the proposed method as PMI is claimed as a main contribution in this paper. If replacing MI with PMI can result in generalizable performance boosting across datasets, the proposed method can be supported with more experimental significance.

Reference:

[1] “**An Empirical Study on Disentanglement of Negative-free Contrastive Learning**”, NeurIPS 2022.

[2] “Challenging Common Assumptions in the Unsupervised Learning of Disentangled Representations.”, ICML 2019

**Questions:**

Please see my comments in previous sectors.

---

> ### Author Response · Authors · 2024-11-21
>
> We thank the reviewer for the insightful and valuable feedback. We explain the concerns point by point below.
>
> **W1: Given that stylized experts make unbalanced trajectory generation in training, it might be helpful to discuss the disentanglement between the stylized code and the state-action pair. Some previous works have built related benchmark for the purpose such as [1] and [2]. Though the benchmarks may not be originally designed for imitation learning, the corresponding metrics might be worth reference. Such evaluation can be adapted by measuring the mutual information or other metrics between the stylized code and the pair.**
>
> R1: After reviewing the methods in [1, 2], we acknowledge that we can adapt $R_{ij}$ and MED to weight the BC loss. However, we still need PMI instead of MI for disentanglement, as our focus is on state-action pairs. Therefore, it is possible to adapt the methods you referred to by replacing MI with PMI in the calculation of $R_{ij}$ and MED. We have included a discussion on the possible improvements through combining our method with other disentanglement methods in the revised version.
>
> [1] “An Empirical Study on Disentanglement of Negative-free Contrastive Learning”, NeurIPS 2022.
>
> [2] “Challenging Common Assumptions in the Unsupervised Learning of Disentangled Representations.”, ICML 2019
>
> **W2: It remains not unclear, at least not generally convincing across datasets, that PMI shows a significant advance than the usual MI for BC. More evidence or discussion about this can be helpful to enhance the significance of the proposed method as PMI is claimed as a main contribution in this paper. If replacing MI with PMI can result in generalizable performance boosting across datasets, the proposed method can be supported with more experimental significance.**
>
> R2: For our experiments, we utilized the Atari and basketball datasets. The Atari dataset represents a complex environment with discrete action spaces in a virtual gaming scenario. In contrast, the basketball dataset is a large, real-world dataset consisting of 500k data points and features a continuous action space. Therefore, we believe that the benchmarks we have chosen demonstrate the generalizability of our method. In future work, we will consider involving other benchmarks (e.g. MuJoCo) to further validate our approach.

---

> ### Author Response · Authors · 2024-11-25
>
> Dear Reviewer ihL7,
>
> Thanks again for your valuable review. We are wondering whether our rebuttal has addressed your concerns. We would love an active discussion with you and hope to address any remaining concern or question you may have.
>
> Best Regards, Authors

---

> > ### Comment · Reviewer_ihL7 · 2024-11-29
> >
> > I appreciate the feedback from the authors. The extra metrics about disentanglement can help understand the nature of the proposed method quantitatively. The authors' discussion and promise about the related experiments would make a beneficial improvement in the next version.
> >
> > On the other hand, the involved benchmarks in the paper still remain limited in my opinion thus not providing significant evidence the generalizability of the proposed method. However, the authors promise to add more benchmarks in the next version. This should help improve the paper with more evidence.
> >
> > I maintain my positive rating for this submission.

---

### Official Review · Reviewer_6kbz · 2024-11-04

**Soundness:** 3
**Presentation:** 3
**Contribution:** 2
**Rating:** 6
**Confidence:** 4

**Summary:**

The paper explores a methodology for deriving diverse policies from expert trajectory data. Rising from the traditional conditional behavior cloning (BC) algorithm, the authors  introduce an additional importance weight based on Pointwise Mutual Information (PMI), to assess the correspondence between state-action pairs and trajectory styles. The experimental results prove the proposed method can improve policy diversity with PMI weighting.

**Strengths:**

1. The topic studied in this paper is quite interesting, with an objective to improve policy diversity.
2. This article clearly defines the problem and provides a detailed exposition of the method.
3. This article undertakes extensive comparative experiments across Circle 2D, Atari games and professional basketball player dataset, which valiate the effectiveness of the proposed algorithm compared with the baseline methods.

**Weaknesses:**

1. This paper aims to enhance policy diversity by incorporating a weighting method based on pointwise mutual information to the Conditional Behavioral Cloning framework. The key proposal of this paper is merely the introduction of a data weighting algorithm, which, in my view, does not represent adequate technical contribution.
2. The manuscript does not include sufficient ablation experiments to validate the efficacy of the proposed data weighting method that utilizes pointwise mutual information. It remains unclear how BC-PMI compares to Conditional Behavioral Cloning that is conditioned on style, where each sample point is assigned equal weight. Additionally, its performance against other traditional data weighting methods has not been adequately compared.

**Questions:**

1. Which behavior cloning (BC) algorithm is selected as the baseline method? Does the proposed PMI general well across different BC algorithms?
2. From Table 1, the improvement of the proposed BC-PMI seems marginal compared with CBC.

---

> ### Author Response · Authors · 2024-11-21
>
> We thank the reviewer for the insightful and valuable feedback. We explain the concerns point by point below.
>
> **W1: This paper aims to enhance policy diversity by incorporating a weighting method based on pointwise mutual information to the Conditional Behavioral Cloning framework. The key proposal of this paper is merely the introduction of a data weighting algorithm, which, in my view, does not represent adequate technical contribution.**
>
> R1: We would like to clarify the contributions of our paper as follows:
>
> 1. For the task of recovering diverse policies from diverse expert data, to the best of our knowledge, we are the first to make the observation that often only a part of a trajectory is highly relevant to the style of a policy. We provide a solid solution (based on pointwise mutual information between state-action pair (s,a) and style variable z) to quantify this, based on which we develop a novel and principled diversity recovering imitation learning method, i.e., BC-PMI.
>
> 2. Our approach is both elegant and effective, because we pay more attention (in a principled way via Pointwise mutual information weighting) to those state-action samples that are more likely generated by a policy with a targeting style, when learning a policy with the style; Moreover, we theorectically unified the BC and CBC as special cases within our framework.
>
> 3. We provide an intuitive motivating example (Circle 2D) and verify the effectiveness of our approach in multiple diversity metrics in the image-based environment Atari and real world basketball datasets.
>
>
> **W2.1: The manuscript does not include sufficient ablation experiments to validate the efficacy of the proposed data weighting method that utilizes pointwise mutual information. It remains unclear how BC-PMI compares to Conditional Behavioral Cloning that is conditioned on style, where each sample point is assigned equal weight.**
>
> R2.1: Thank you for highlighting the need for additional ablation experiments to further validate the efficacy of our proposed data weighting method.
> We would like to point out that we have provided a comparison between BC-PMI and CBC in both the Atari and basketball benchmarks (please refer to Table 3 and Table 4), and we provides more ablation experiments in Table 8 and Table 9 as suggested by reviewers.
> Additionally, we wish to emphasize that Conditional Behavioral Cloning (CBC) can be viewed as a special case of our BC-PMI method. In CBC, each state-action pair (s, a) is assigned an equal weight of 1 across all style labels, which means it does not differentiate between the relevance of different samples to the style being learned. In contrast, our PMI method adjusts the weights according to the specific relevance of each sample to the style, thereby enhancing the learning process by focusing on the most informative samples. We further explain this in Fig 7 in Appendix C.
>
> **W2.2: Additionally, its performance against other traditional data weighting methods has not been adequately compared.**
>
> R2.2: Due to time limited, we implement DWBC [1] as our competitor. (There are also other candidates such as [2]). For the DWBC method, the original DWBC algorithm uses $d(s, a, log \pi)$ to determine whether the current sample is generated by the expert policy. In our setting, we use $d(s, a, z)$ to determine
> whether the current sample belongs to style z. The experiment results is shown in Table 9. We can see that DWBC performes better than BC but worse than BC-PMI (Ours). We have add the results in Appendix E.
>
> [1] Xu, Haoran, et al. "Discriminator-weighted offline imitation learning from suboptimal demonstrations." International Conference on Machine Learning. PMLR, 2022.
>
> [2] Sasaki, Fumihiro, and Ryota Yamashina. "Behavioral cloning from noisy demonstrations." International Conference on Learning Representations. 2020.

---

> > ### Author Response · Authors · 2024-11-21
> >
> > **Q1: Which behavior cloning (BC) algorithm is selected as the baseline method? Does the proposed PMI general well across different BC algorithms?**
> >
> > A1: We selected vanilla BC (Equation 1) as our baseline method. We chose this fundamental approach to validate our concept effectively, as it is simple to implement and allows for straightforward comparison.
> >
> > We have also experimented with combining PMI with CGAIL, as shown in Table 9. This combination brings improvement through PMI weighting on CGAIL (compared to CGAIL alone). However, the method suffers from unstable training (a challenge also noted in the GAIL method) and is difficult to tune (within such a short rebuttal time) to achieve a comparable score to BC-PMI.
> >
> > **Q2: From Table 1, the improvement of the proposed BC-PMI seems marginal compared with CBC.**
> >
> > A2: In fact, Table 1 primarily features a toy example to visually present our motivation to readers. Due to the simplicity of this environment, it is challenging to highlight the differences between the algorithms. Therefore, we have subsequently chosen the Atari environment, based on images and human data, as well as the basketball environment, which includes a large amount of human data, for comparison. We have also provided other comparison and ablation to further demonstrate the effectivity of BC-PMI in Appendix E.

---

> ### Author Response · Authors · 2024-11-25
>
> Dear Reviewer 6kbz,
>
> Thanks again for your valuable review. We are wondering whether our rebuttal has addressed your concerns. We would love an active discussion with you and hope to address any remaining concern or question you may have.
>
> Best Regards, Authors

---

> ### Comment · Reviewer_6kbz · 2024-11-27
> **Response**
>
> Thank you for the detailed explanations and significant effort in addressing my concerns, which have resolved most of my points.  I encourage the authors to validate the effectiveness and applicability of PMI on a wider range of advanced BC policies when time permits in the future. Considering the overall improvements, I have decided to increase my rating.

---

> > ### Author Response · Authors · 2024-11-28
> > **Official Comment by Authors**
> >
> > Thank you very much for your thoughtful feedback. Your insights are invaluable in enhancing the quality of our work. We appreciate your suggestion to validate the effectiveness and applicability of PMI across a broader range of advanced BC policies, as this is indeed an important direction for future research.

---

### Author Response · Authors · 2024-11-21

We would like to express our sincere gratitude for the thorough review and valuable feedback on our paper. The reviewers' insights and suggestions are quite helpful to improve the quality and clarity of our work.

We are encouraged the topic is interesting (Reviewer 6kbz) and the method is well-motivated (Reviewer ihL7, Reviewer BUhV). The paper is well-written and clearly communicates the key ideas (Reviewer 6kbz, fKFr, BUhV). The method is validated via extensive experiments (Reviewer 6kbz, BUhV)

We have uploaded a revised version of the paper, with changes highlighted in blue within the main text. Additional analyses and experiments have been included in the Appendix. The main revisions covered in the Appendix are as follows:
- Added more detailed comparison and explanation of the weights of BC-PMI and CBC (Appendix C)
- Added more detailed explanation and experiments of the label function (Appendix D)
- Added more comparison of the calibration with more number of styles (Appendix E.1)
- Added more ablation studies, comparing BC-PMI to BC-classifier, a weighted BC competitor and PMI+CGAIL.(Appendix E.2)

Please see the revision paper for more details.

---

### Meta-Review · Area_Chair_ZEvs · 2024-12-20

**Metareview:**

Summary: This paper proposes Behavioral Cloning with Pointwise Mutual Information Weighting (BC-PMI), a method that enhances policy diversity by weighting state-action pairs based on their relevance to trajectory styles using Pointwise Mutual Information (PMI). Experimental results across three settings, Circle 2D, Atari Games, and the professional basketball datasets, show that BC-PMI outperforms baselines in terms of style calibration, while also providing insights into the smooth interpolation between traditional behavioral cloning and clustering-based behavior cloning.


Strengths and Weaknesses:

Generally, the reviewers find the topic compelling, the problem clearly defined, and the method well-explained, with an intuitive motivation that addresses the shortcomings of existing alternatives and strong results across multiple datasets, making the paper a significant contribution to improving the efficiency of imitation learning.

While the paper presents a promising approach, several weaknesses were identified by the reviewers. The original manuscript lacks several key ablation experiments and comparisons with other reweighting methods or style-conditioned BC, as well as evaluations on state and action diversity, which are necessary to fully establish the superiority of the proposed method. There were also concerns regarding the fairness of the comparisons due to differing types of supervision. Additionally, the experimental section is brief and does not analyze the method's performance as the number of styles increases or in the presence of noisy labels, which is particularly important given the reliance on potentially imperfect labels.

During the discussion phase, the authors' responses, along with the additional experimental results, effectively address the reviewers' concerns.

All reviewers recommend acceptance, and ACs support their recommendation. The authors should address the main points raised in the reviews, particularly the additional analyses and experiments, such as validating the effectiveness and applicability of PMI on a broader range of advanced BC policies and conducting more comprehensive comparisons with other data weighting methods, as promised in the rebuttal, when preparing the camera-ready version.

**Additional Comments On Reviewer Discussion:**

The current recommendation is based on the reviewers’ comments and the outcome of the author-reviewer discussion. The authors should address the main points raised in the reviews, particularly the additional analyses and experiments promised in the rebuttal, when preparing the camera-ready version.

---

### Decision · Program_Chairs · 2025-01-22

Accept (Poster)